# OpenReview forum: "Retrieval-augmented Adaptive Decoding for Improving Truthfulness in Open-ended Generation"
_TMLR — Rejected by TMLR_

### Review · Reviewer_tgXq · 2025-12-14

**Summary Of Contributions:**

The paper introduces Retrieval-Augmented Decoding (RAD), a decoding-time method designed to improve truthfulness in LLMs without applying retraining. The method proposes to construct a grounding space from a few annotated examples, storing context embeddings and next-token logits from ground-truth responses. During inference, RAD retrieves similar contexts and aggregates their logits to adjust the model’s predictions. Experiments across three benchmarks (TruthfulQA, WikiQA, Alpaca) and four LLMs show consistent improvements in terms of truthfulness and informativeness over different baselines.

Strengths
- In contrast to fine-tuning methods, which require large datasets, and in-context learning, which suffers from prompt sensitivity and reduced robustness across domains, RAD introduces a lightweight, logit-level intervention that avoids retraining and multi-pass generation, making it computationally efficient. Moreover, the approach is conceptually simple and sample-efficient, achieving strong results with as few as 10 annotated examples.
- The paper presents a comprehensive experimental evaluation, including experiments across multiple datasets and models and compares against strong baselines (DoLa, CAD, ID, KATE) with improvements in truthfulness and informativeness.
- The method is shown to be robust across domains and out-of-distribution settings, outperforming prompt-based methods like KATE.
- Finally, the presentation is of good quality: the methodology is well-structured (including an algorithmic description and illustrative figures), the language is easy to follow, and the ablation studies on hyperparameters are detailed (chunk size, threshold, interpolation weight).

Weaknesses
- The major weakness is that the method is similar to semi-parametric LLM methods, especially kNN-LM (Khandelwal, U., Levy, O., Jurafsky, D., Zettlemoyer, L., & Lewis, M. Generalization through Memorization: Nearest Neighbor Language Models. In International Conference on Learning Representations), which augments a pre-trained LLM by linearly interpolating its next-token distribution with that from k nearest neighbors in a datastore. However, the manuscript does not cite it, nor does it explain the similarities/differences, nor does it compare against the kNN-LM method.
- The paper relies on truthfulness and informativeness scores from an external LLM acting as a judge (Cohere API). No human evaluation, no correlation analysis between the judge and human judgments, and no robustness analysis across different judges is presented. It is therefore not clear whether the obtained results are potentially attributed to error or bias from the selected LLM judge.
- I wonder why performance sometimes drops when larger grounding spaces are considered (see Figure 3). This seems counterintuitive and is not well-explained in the paper.
- Moreover, although the paper claims computational efficiency, it does not provide quantitative comparisons in terms of decoding time or computation (e.g., FLOPs, latency, or throughput) against baselines.
- Last, while the paper focuses on QA benchmarks, evaluation on more complex tasks (e.g., multi-hop reasoning, dialogue) could demonstrate broader applicability and limitations.

**Audience:**

Yes

**Audience Explanation:**

The paper addresses a topic of interest: improving truthfulness of LLMs via decoding-time interventions. There is ongoing work on decoding strategies (e.g., contrastive decoding, DoLa), semi-parametric LMs, and factuality control, and RAD fits squarely within this line of research. The idea of using a small, curated grounding space to steer logits towards truthful behavior, without retraining or multi-pass generation, is practically appealing, especially in scenarios where only limited supervision is available, and model weights cannot be modified.

**Broader Impact Concerns:**

I do not have specific ethical concerns beyond those commonly associated with work on factuality and truthfulness in LLMs. The method aims to reduce hallucinations and improve truthful behavior, which is generally beneficial. However, two points could be briefly addressed:

1. Since the paper relies on an external LLM judge to assess truthfulness and informativeness, it would be useful to acknowledge that such judges may themselves be biased or factually unreliable in some cases, and that improved truthfulness under this metric does not automatically guarantee alignment with human notions of correctness.

2. A method that steers model outputs could be misused to steer models toward particular narratives or preferences. A brief discussion highlighting that RAD can, in principle, be used both to improve and to manipulate model outputs, and that responsible use requires a careful choice of grounding data and evaluation, would be appropriate.

**Claims And Evidence:**

Yes

**Claims Explanation:**

The empirical section is generally solid: the authors evaluate RAD on three benchmarks (TruthfulQA, WikiQA, Alpaca) and four LLMs, and compare against several decoding baselines (DoLa, CAD, ID, KATE). The reported gains in truthfulness and informativeness are consistent across models and datasets, and the paper includes ablations on key hyperparameters (grounding size, chunk size, similarity threshold, interpolation weight), which supports the claims of sample efficiency and robustness.
However, there are some issues:
1. The evaluation relies entirely on an external LLM judge (Cohere) for truthfulness and informativeness scores. While this is aligned with recent practice, there is no human evaluation or analysis of judge reliability and bias, so the strength of the claims ultimately depends on the correctness of this single judge.
2. The claim of computational efficiency is plausible given the design (single-pass decoding with retrieval), but it is not backed by quantitative runtime or resource comparisons against baselines.

**Requested Changes:**

1. The method is very similar in spirit and mechanics to kNN-LM and related semi-parametric LM approaches, which interpolate the base LM’s next-token distribution with a retrieval-based distribution from a datastore. The authors should cite this line of work (e.g., Khandelwal et al., Generalization through Memorization: Nearest Neighbor Language Models), explain the similarities and differences between RAD and kNN-LM–style methods, and ideally, include a comparison against a simple kNN-LM–style baseline adapted to this setting.

2. Since all key metrics are obtained from a single LLM judge (Cohere API), it is advised to either provide a small human evaluation on a subset of examples to verify that the LLM judge’s labels correlate reasonably with human judgments, o use at least one additional LLM judge and report agreement / robustness across judges.

3. Figure 3 shows that performance can decrease when the grounding space becomes larger, which is counterintuitive. The paper should provide a more detailed analysis or explanation of why this happens and how to choose an appropriate grounding size in practice.

4. Explain how the examples used to build the grounding space are selected. Are they chosen randomly or using any heuristic for quality/representativeness? Since the method is sample-efficient, the selection method can have a substantial impact on performance.

5. Please add runtime or throughput comparisons (e.g., tokens per second or wall-clock time) for RAD vs the other baselines and a brief discussion of memory/storage overhead for the grounding space (embeddings + logits) and retrieval.

---

> ### Author Response · Authors · 2026-01-05
>
> Thank you for your constructive feedback. We sincerely appreciate your positive assessment that our method is `computationally efficient` and `sample-efficient`. We are also glad that you found our paper is `clearly written` and `easy to follow`, with a `comprehensive experimental evaluation` and a `well-structured methodology`. Below, we address your concerns in detail. We have revised the paper accordingly.
>
> **W1: The major weakness is that the method is similar to semi-parametric LLM methods, especially kNN-LM (Khandelwal, U., ICLR 2020), which augments a pre-trained LLM by linearly interpolating its next-token distribution with that from k nearest neighbors in a datastore. However, the manuscript does not cite it, nor does it explain the similarities/differences, nor does it compare against the kNN-LM method.**
>
> Following your suggestion, we have added experimental comparisons with kNN-LM and clarified its relationship with RAD in Sections 4.1 and 4.2. kNN-LM interpolates LM predictions with a kNN distribution computed using L2 distances in embedding space over a large datastore built from millions of reference instances (data-intensive), retaining only a single next-token prediction per context, which often leads to unstable gains. In contrast, RAD aggregates full logit distributions from retrieved evidence, enabling more stable and fine-grained decoding adjustments and yielding consistent improvements across tasks and model scales (Tables 1, 2). By further decomposing evidence into evenly sized chunks and filtering low-quality contexts, RAD mitigates retrieval noise and preserves richer distributional signals, resulting in stronger out-of-distribution generalization (Table 3).
>
> **W2: The paper relies on truthfulness and informativeness scores from an external LLM acting as a judge (Cohere API). No human evaluation, no correlation analysis between the judge and human judgments, and no robustness analysis across different judges is presented. It is therefore not clear whether the obtained results are potentially attributed to error or bias from the selected LLM judge.**
>
> Thank you for raising this concern. We have added results using an additional proprietary judge (Gemini API) as well as open-source automatic metrics (ROUGE-L and BERTScore) to demonstrate strong agreement across different evaluation protocols (Section 4.2 and Appendix A.2).
>
> **W3: I wonder why performance sometimes drops when larger grounding spaces are considered (see Figure 3). This seems counterintuitive and is not well-explained in the paper.**
>
> Thank you for this observation. We have added a clarification below Figure 3 in Section 4.3. Using 400 samples provides little benefit and can slightly degrade performance on specific models. Even with semantically retrieved contexts, pairwise cosine similarity can be low, and a grounding space of N=400 is still small compared to the millions used in kNN-LM (Khandelwal et al., 2020), so adding more candidates at this scale may introduce noisier or conflicting signals during aggregation.
>
> **W4: Moreover, although the paper claims computational efficiency, it does not provide quantitative comparisons in terms of decoding time or computation (e.g., FLOPs, latency, or throughput) against baselines.**
>
> Thank you for this suggestion. We have included a quantitative latency analysis in Section 4.3 (Latency Analysis paragraph). Grounding space construction takes less than 30 minutes for most datasets, and retrieval adds approximately 4.6–4.9 ms per step. For typical outputs of 2–3 sentences (~200 tokens), this results in only 1 second of additional decoding time compared to other logit-modifying decoding methods.
>
> **W5: Last, while the paper focuses on QA benchmarks, evaluation on more complex tasks (e.g., multi-hop reasoning, dialogue) could demonstrate broader applicability and limitations.**
>
> Thank you for mentioning this. We have added experiments on HaluEval (covering dialogue and summarization tasks) in Section 4.2 (Dialogue and Summarization Tasks paragraph). Across all settings (Table 2), RAD achieves the strongest average performance, outperforming decoding and prompting baselines on both tasks. Gains are especially notable on Qwen2.5-3B (up to 5.8% over Greedy), while prompting baselines like KATE provide little help, showing that RAD is effective for smaller, hallucination-prone models and generalizes beyond QA.
>
> **C1: Explain how the examples used to build the grounding space are selected. Are they chosen randomly or using any heuristic for quality/representativeness? Since the method is sample-efficient, the selection method can have a substantial impact on performance.**
>
> Thank you for mentioning this. By decomposing evidence into evenly sized chunks and filtering low-quality contexts, RAD reduces the sensitivity to data selection. For all experiments, we use the first 100 instances from training sets, which is now clarified in Appendix A.1.

---

### Review · Reviewer_RsQN · 2025-12-21

**Summary Of Contributions:**

This paper proposes a lightweight decoding-time method to improve the truthfulness of LLM outputs without model training, which is called Retrieval-Augmented Decoding (RAD). The method builds a retrieval corpus to find the most similar truthful examples to adjust the logits of tokens in the base LLM.  Specifically, according to the context chunk embedding and a small set of truthful examples, RAD retrieves semantically similar context chunks from the retrieval corpus at each decoding step and adds the aggregated logits to the base model’s logits. The authors evaluate RAD on several open-ended QA and instruction-following benchmarks and the experimental results outperform compared with other training-free in-context learning methods.

Strengths:
1. The RAD method is simple, implementable, and training-free.
2. The experimental results show some evidence of cross-task generalization (mainly in TruthfulQA and WikiQA).

Weaknesses:
1. The paper is motivated as addressing hallucinations, but the evaluation protocal and metrics are TruthfulQA-style "truthfulness + informativeness" scorings. The connection to general hallucination settings is not clearly explaned.
2. Potential overlap / leakage concerns are not analyzed. The retrieval corpus is built from a small set of examples drawn from the dataset’s training split (such as 100 samples), so improvements on in-distribution tests could be influenced by entity/phrase overlap or template duplication.  This paper does not quantify overlap or run overlap-controlled experiments. However, the method may rely heavily on overlap effects, making it difficult to verify whether the gains come from genuine semantic transfer or near-duplicate/template matching.
3. Inference efficiency is not quantified. The method performs retrieval and aggregation at every decoding step, but the paper does not report runtime overhead. The paper does not clarify whether the inference overhead is acceptable.

**Audience:**

Yes

**Audience Explanation:**

Readers interested in reliable retrieval-augmented inference without model training would find the idea and empirical results informative. The method is simple. It may be a complement to the retrieval based augmented methods in certain scenarios.

**Claims And Evidence:**

No

**Claims Explanation:**

1. The problem definition and proposed solution is mismatched. The paper is motivated as addressing hallucinations, but the evaluation protocal and metrics are TruthfulQA-style "truthfulness + informativeness" scorings.

2.  The paper evaluates only on TruthfulQA, WikiQA, and Alpaca, without adequately explaining why these three benchmarks represent key distributions of hallucinations. There is no typical hallucination benchmarks in experiments, raising concerns about potential selection bias (cherry-picking).

3. There is a risk of information leakage. There may be "nearest-neighbor answer carrying" or "template leakage" rather than genuine factual correction. RAD retrieves the most similar chunks and adjust the logits of tokens in the base LLM by these chunkes' logits. This can yield apparent improvements on some datasets (e.g. TruthfulQA/WikiQA), but does not necessarily indicate a true improvement in factual reasoning. Instead, it may reflect distribution similarity and "answer/template exploitation". The paper does not analyze phrase overlap between the support set and the test set. Under the paper’s settings, in-distribution gains are not surprising and the key question is whether substantial overlap leads to leakage of answer patterns.

4. There are incomplete explanations of case study. In Table 4, the WikiQA example, RAD changes the year from 2023 to 2008, but the paper does not show the specific retrieved chunks, similarity weights, or the corresponding logit contributions. As a result, it is unclear whether the correction comes from capability transfer, or whether the same fact / template brings leakage the answer.

**Requested Changes:**

1. The problem definition and proposed solution is mismatched. The paper is motivated as addressing hallucinations, but the evaluation protocal and metrics are TruthfulQA-style "truthfulness + informativeness" scorings.

    I suggest that the authors clearly define the "hallucination" mentioned in the paper and use it consistently throughout the paper (either by narrowing the scope to truthfulness in open-ended QA, or by incorporating a more general settings and experiments), to avoid overgeneralizing the concept and overstating the generality of the conclusions.

2. The paper evaluates only on TruthfulQA, WikiQA, and Alpaca, without adequately explaining why these three benchmarks represent key distributions of hallucinations. There is no typical hallucination benchmarks in experiments, raising concerns about potential selection bias (cherry-picking).

    I suggest that the authors add at least one or two more typical hallucination evaluation benchmarks to alleviate concerns about cherry-picking.

3. There is a risk of information leakage. There may be "nearest-neighbor answer carrying" or "template leakage" rather than genuine factual correction. RAD retrieves the most similar chunks and adjust the logits of tokens in the base LLM by these chunkes' logits. This can yield apparent improvements on some datasets (e.g. TruthfulQA/WikiQA), but does not necessarily indicate a true improvement in factual reasoning. Instead, it may reflect distribution similarity and "answer/template exploitation". The paper does not analyze phrase overlap between the support set and the test set. Under the paper’s settings, in-distribution gains are not surprising and the key question is whether substantial overlap leads to leakage of answer patterns.

    I suggest that the authors display phrase overlap between the retrieved chunks and the test cases and run deduplication experiments to demonstrate that the gains are not driven by nearest neighbors leakage answer patterns.

4. Incomplete Explanation of Case Study.

    1）The Table 4 case study is not fully explaned. In Table 4, the WikiQA example, RAD changes the year from 2023 to 2008, but the paper does not show the specific retrieved chunks, similarity weights, or the corresponding logit contributions. As a result, it is unclear whether the correction comes from capability transfer, or whether the same fact / template brings leakage the answer.

   I suggest that the authors show the top retrieved neighbor chunks used by RAD (including the text, similarity scores, and weights), as well as the logit increments on the key tokens, to make the method’s contribution clearer.

    2）The mechanism in explaining "KATE fails to retrieve relevant exemplars but RAD still corrects the answer" in case study is unclear.  If KATE cannot retrieve relevant exemplars while RAD can correct the output, the paper should provide evidence (e.g., retrieved neighbors and their weights) to explain this discrepancy (e.g. whether RAD is matching phrase-level templates).

5. Evaluation Reliability. The evaluation relies on a single LLM judge, without reporting multi-judge agreement or any human verification, which raises concerns about robustness and reproducibility.

    I suggest adding at least one supplementary experiments mentioned above to strengthen the persuasiveness of the experimental results.

6. There is no comparison to standard RAG. The method is retrieval-augmented, so why is there no comparison to a standard RAG baseline? The paper uses KATE as a retrieval-augmented baseline, but KATE is exemplar-based in-context learning and cannot substitute for evidence-based RAG. Especially for knowledge QA tasks like WikiQA, the lack of a standard RAG comparison makes it difficult to position RAD’s practical advantages and applicability boundaries. What is RAD’s advantage over standard RAG?

    I suggest including a standard RAG baseline and clearly delineating RAD’s advantages and applicability boundary relative to RAG (e.g., shorter context length, reduced prompt sensitivity, etc.).

7. Writing suggestions.

    1）Inference efficiency is a concern. The paper states it is "retrieving and aggregating highly relevant contexts at each decoding step". Would this be slow at decoding time? Would it be better to apply this auxiliary decoding only at key steps? The paper does not clarify whether the inference overhead is acceptable or prohibitive.

    2）Imprecise Terminology. The paper uses "training samples" to describe the data used to construct the retrieval corpus, but this is not data used for parameter training. The term "training samples/data" is potentially misleading.

---

> ### Author Response · Authors · 2026-01-05
>
> Thank you for your constructive feedback. We are glad that you found our method is `simple`, `implementable`, `training-free`, and capable of `cross-task generalization`. Below, we address your concerns in detail. We have revised the paper accordingly.
>
> **W1: The paper is motivated as addressing hallucinations, but the evaluation protocal and metrics are TruthfulQA-style "truthfulness + informativeness" scorings. The connection to general hallucination settings is not clearly explaned.**
>
> Our work focuses on improving truthfulness and factuality in open-ended generation. In the revised version, we have clarified this motivation and connection in the title, introduction, related work, and experiments sections.
>
> **W2: The paper evaluates only on TruthfulQA, WikiQA, and Alpaca, without adequately explaining why these three benchmarks represent key distributions of hallucinations. There is no typical hallucination benchmarks in experiments, raising concerns about potential selection bias (cherry-picking).**
>
> Thank you for this concern. We have extended our experiments to include HaluEval, a widely used hallucination-oriented benchmark covering dialogue and summarization, to better demonstrate robustness across hallucination settings (Table 2, Section 4.2). The results show that RAD consistently outperforms existing decoding and prompting baselines across both dialogue and summarization tasks, with especially strong gains on smaller, hallucination-prone models, confirming that RAD generalizes robustly beyond QA.
>
> **W3 (information leakage concern):**
>
> This is an insightful concern. We have added an analysis in Section 4.3 (Effect of Exact Match paragraph) showing that while RAD benefits from exact-match contexts when available, it does not rely on them exclusively and achieves strong performance by leveraging semantic similarity even in their absence.
>
> **W4: There are incomplete explanations of case study. In Table 4, the WikiQA example, RAD changes the year from 2023 to 2008, but the paper does not show the specific retrieved chunks, similarity weights, or the corresponding logit contributions. As a result, it is unclear whether the correction comes from capability transfer, or whether the same fact / template brings leakage the answer.**
>
> Thank you for this feedback. Due to space constraints, we have expanded the case study and moved detailed retrieved chunks, similarity weights, and logit contributions to the appendix (Table 10, Appendix A.6). This example also illustrates how semantically similar (but non-overlapping) contexts steer the generation toward the correct answer.
>
> **C1: Evaluation Reliability. The evaluation relies on a single LLM judge, without reporting multi-judge agreement or any human verification, which raises concerns about robustness and reproducibility. I suggest adding at least one supplementary experiments mentioned above to strengthen the persuasiveness of the experimental results.**
>
> Thank you for this suggestion. We have added results using Gemini API and open-source metrics (ROUGE-L, BERTScore) to confirm consistency across judges (Section 4.2 and Appendix A.2).
>
> **C2 (comparision to standard RAG baseline):**
>
> Thank you for this important point. We have included kNN-LM (Khandelwal et al., Generalization through Memorization: Nearest Neighbor Language Models, ICLR 2020) a retrieval-augmented non-prompting baseline, and discussed the similarities, differences, and respective advantages of RAD over both kNN-LM and traditional prompt-based RAG in Sections 4.1 and 4.2.
>
> **C3: Writing suggestions.**
>
> **1) Inference efficiency is a concern. The paper states it is "retrieving and aggregating highly relevant contexts at each decoding step". Would this be slow at decoding time? Would it be better to apply this auxiliary decoding only at key steps? The paper does not clarify whether the inference overhead is acceptable or prohibitive.**
>
> **2) Imprecise Terminology. The paper uses "training samples" to describe the data used to construct the retrieval corpus, but this is not data used for parameter training. The term "training samples/data" is potentially misleading.**
>
> Thank you for these helpful suggestions.
> - We have included a quantitative latency analysis in Section 4.3 (Latency Analysis paragraph). Grounding space construction takes less than 30 minutes for most datasets, and retrieval adds approximately 4.6–4.9 ms per step. For typical outputs of 2–3 sentences (~200 tokens), this results in roughly 1 second of additional decoding time compared to other logit-modifying decoding methods. We agree that selectively applying retrieval at high-uncertainty steps is promising but requires reliable uncertainty estimation; we leave this for future work.
> - We have replaced potentially misleading terms like “training samples” with “reference instances” or “reference set” throughout the paper to clarify that this data is used only for non-parametric retrieval.

---

> > ### Comment · Reviewer_RsQN · 2026-01-22
> >
> > We thank the authors for their detailed responses. We appreciate the clarifications and additional experiments provided in the revised version. Below, we respond point by point to the authors’ replies, indicating which concerns are addressed and which remain unresolved.
> >
> > The responses to W1, W2, C1, and C3 adequately address my concerns. However, several key questions remain unanswered.
> >
> > W3: This analysis does not fully address the information leakage concern. Removing exact matches does not eliminate subtler leakage from semantically similar matches, template reuse, or low-entropy factual slots (e.g., years or entity attributes). Since RAD relies on semantic similarity and directly adjusts token-level logits, its gains on in-distribution benchmarks may be difficult to distinguish from nearest-neighbor answer copying or distributional bias amplification.
> >
> > W4: The case study does not resolve the core concern. This case study mainly shows token-level logit bias (e.g., selecting “0” versus “2” in a year token) rather than improved factual understanding. Since the correction occurs at a low-entropy, template-driven slot, the result remains consistent with a nearest-neighbor explanation, leaving it unclear whether the gain reflects real capability improvement or distributional exploitation.
> >
> > C2: Although the RAD is closer to kNN-LM, from a functional perspective RAD behaves like a RAG method that aims to correct factual outputs using retrieved external contexts. From this viewpoint, comparisons with standard RAG methods and evaluations of evidence grounding are important. Comparing mainly to kNN-LM avoids these questions, but does not clarify whether RAD improves factual reasoning or mainly performs distributional correction through similarity-based retrieval.
> >
> > Overall: While RAD appears effective in certain in-distribution settings, the current evaluation does not convincingly demonstrate that it improves general factual reasoning or robustness beyond nearest-neighbor–style distribution correction. Therefore, I do not find the current version fully satisfactory.

---

> > > ### Author Response · Authors · 2026-01-24
> > >
> > > We thank the reviewer for the thoughtful follow-up comments. Below we clarify the remaining concerns and describe concrete actions taken to improve clarity and scope alignment.
> > >
> > > **- W3: Information leakage / nearest-neighbor concern**
> > >
> > > We agree that RAD explicitly leverages distributional similarity across contexts. This behavior is **intentional and central** to the method, rather than an unintended leakage artifact.
> > >
> > > RAD does **not** aim to improve factual reasoning, induce new knowledge, or extrapolate beyond the training distribution. Instead, it targets **decoding-time distribution calibration**, based on the intuition that *similar conditional contexts induce similar next-token distributions*. At each step, RAD combines the base model logits with the retrieved logits using a weighting factor $\alpha$ (e.g. $\alpha$=0.5, Eq. 8) to control the influence of the retrieved signals, while still fully leveraging the base model's original logits. From this perspective, concerns about "distributional exploitation" or "nearest-neighbor effects" describe the **mechanism of RAD**, not a failure mode.
> > >
> > > Importantly, RAD differs from nearest-neighbor answer copying because it does not retrieve or inject answers, templates, or evidence at the input level. Instead, it aggregates token-level logits from multiple similar contexts, yielding a **soft, distributional bias** rather than deterministic copying or memorization.
> > >
> > > **- W4: Case study and "low-entropy slot" interpretation**
> > >
> > > We respectfully disagree with the characterization of the corrected tokens as "low-entropy slots" from a decoding perspective. While such slots are low-entropy in the ground-truth distribution, they are often high-entropy under the model's conditional distribution, with multiple candidate tokens receiving similar logits and poorly calibrated probabilities. This phenomenon, illustrated in the qualitative example in Appendix A.5, is also observed in other work on next-token calibration (Section 4.2 [1]) and entropy calibration of language models (Section 4 [2]).
> > >
> > > RAD is designed to operate precisely in this regime: when competing tokens receive similar probabilities, aggregating distributions from similar contexts reallocates probability mass toward empirically supported modes. The observed corrections therefore reflect **decoding-time calibration**, not improved factual understanding or symbolic reasoning.
> > >
> > > **- C2: Relation to RAG and factual reasoning**
> > >
> > > We agree that RAD can be viewed as retrieval-augmented in a broad sense. However, its objective and operating level differ fundamentally from standard RAG. Standard RAG improves factual correctness via **external evidence grounding at the input level**. In contrast, RAD operates **without evidence injection, supervision, or prompt modification**, and instead performs **output-level distribution reshaping**.
> > >
> > > As such, RAD does not claim to improve factual reasoning or evidence attribution. Its contribution is to show that non-parametric, retrieval-based distribution regularization at decoding time can substantially improve factual outcomes under fixed models, particularly in uncertainty-prone regimes.
> > >
> > > **Overall clarification of scope**
> > >
> > > To avoid ambiguity, we emphasize that RAD should be interpreted as a **decoding and calibration method**, not a reasoning or knowledge acquisition method. Its gains stem from exploiting **conditional distribution smoothness** in context space, and its effectiveness in in-distribution settings is expected and aligned with its design goals.
> > >
> > > [1] Lovering et al., Language Model Probabilities are Not Calibrated in Numeric Contexts (ACL 2025).
> > >
> > > [2] Cao et al., On the Entropy Calibration of Language Models (NeurIPS 2025).

---

> > > > ### Author Response · Authors · 2026-01-24
> > > >
> > > > **Action Plan**
> > > >
> > > > Based on the reviewer’s feedback, we will clarify the scope and intent of RAD in the Abstract, Introduction, and Method sections to avoid potential ambiguity.
> > > >
> > > > - **Abstract**:
> > > >
> > > > We will clarify that RAD does not aim to improve factual reasoning or knowledge acquisition. Instead, it performs decoding-time distribution calibration by exploiting the smoothness of conditional next-token distributions across similar contexts.
> > > >
> > > > - **Introduction**:
> > > >
> > > > We will explicitly state that RAD does not target symbolic reasoning, or model knowledge expansion. Rather, it improves truthfulness by calibrating the model's output distribution at decoding time, through retrieving and aggregating next-token logits from highly similar contexts. Importantly, this aggregation is controlled via a weighting factor, ensuring that retrieved logits act as a regularization signal rather than overriding the base model's predictions.
> > > >
> > > > - **Method**:
> > > >
> > > > We will clarify that semantic similarity is used solely as an efficient proxy for identifying contexts with similar conditional token distributions. RAD does not assume semantic transfer, exemplar-based reasoning, or answer copying, and operates purely at the level of probabilistic logit aggregation.
> > > >
> > > > ---
> > > > These clarifications do not alter the core method, assumptions, or experimental results, and are intended solely to improve conceptual clarity without affecting the contributions evaluated by other reviewers.
> > > >
> > > > We thank the reviewer for helping us sharpen the conceptual framing and improve the clarity of the paper.

---

### Review · Reviewer_Lg6E · 2025-12-22

**Summary Of Contributions:**

The paper introduces Retrieval Augmented Decoding (RAD), a decoding strategy that reduces hallucinations. RAD first constructs a grounding space consisting of a small list of ground truths (GT), embed them along with the next token logits. At test time, semantically meaningful GTs are retrieved and their corresponding next token logits are used to adjust the generated logits to form a truth-aligned logits.

Strengths:
1. The proposed framework is model agnostic and easy to understand.
2. Benchmark results on a variety of open source datasets including WikiQA, Alpaca, shows its superior performance.

Weaknesses:
1. RAD lack real-world use cases since it's infeasible to construct a grounding space for general purpose use cases.
2. Test time retrieval could significant increase the computational overhead at inference-time which limit real-world  applicability. The ideal paradigm would be moving as much computational work to train time as possible.

**Audience:**

No

**Audience Explanation:**

1. RAD lacks practical relevance to real-world LLMs which are exaodiuarily generalizable which make it not suitable for audience of TMLR.
2. Findings are only benchmarked on selected datasets such as Alpaca, which are limited to support the claim made in the paper.

**Claims And Evidence:**

No

**Claims Explanation:**

1. Even though a few datasets are benchmarked to verify the performance of RAD, they still fail to back the claim that open-ended question-answering has been improved to reduce hallucinations.

**Requested Changes:**

1. Improve the scope of the benchmarking to show RAD can in fact reduce hallucination for general-purpose language models.
2. If not the above is not doable, I'd recommend adjusting the claim to focus on domain specific findings.

---

> ### Author Response · Authors · 2026-01-05
>
> Thank you for your constructive feedback. We are glad that you found our method is `easy to understand` and that it demonstrates `superior performance` across various datasets. Below, we address your concerns in detail. We have revised the paper accordingly.
>
> **W1: RAD lack real-world use cases since it's infeasible to construct a grounding space for general purpose use cases.**
>
> Thank you for raising this concern. Our method exhibits strong out-of-distribution generalization and remarkable sample efficiency, achieving competitive performance with as few as 10 reference instances. This allows the grounding space to be constructed from small, high-quality, and easily accessible corpora (e.g., recent Wikipedia dumps or domain-specific datasets), making it feasible for many real-world open-ended generation scenarios where limited curated reference data is available.
>
> **W2: Test time retrieval could significant increase the computational overhead at inference-time which limit real-world applicability. The ideal paradigm would be moving as much computational work to train time as possible.**
>
> Thank you for this feedback. We agree that shifting computation to training time is ideal when possible. However, test-time retrieval offers a lightweight, training-free alternative that excels in both computational and data efficiency. In our experiments, retrieval adds only about 4.6–4.9 ms per decoding step (reported in Section 4.3, Latency Analysis), resulting in approximately 1 second of additional inference time for typical outputs of ~200 tokens. This overhead is substantially lower than that of training-based methods (e.g., SFT or RLHF, which require hours to days of training) while delivering comparable or superior factuality improvements.

---

### Author Response · Authors · 2026-01-05

We sincerely thank all reviewers for their time and valuable feedback. We are encouraged that the reviewers appreciate the **clarity and structure of our paper**, the **simplicity and practicality** of our **training-free decoding method**, and its **strong empirical performance**, **computational efficiency**, and **sample efficiency**, particularly in settings with limited supervision [*tgXq, RsQN, Lg6E*].

In response to the raised concerns, we have made the following major revisions:
- Added comparisons with kNN-LM, discussions of similarities/differences with our method, and analyses addressing potential information leakage and exact-match reliance [*tgXq, RsQN*].
- Extended experiments to HaluEval (dialogue and summarization), strengthened evaluation with additional judges (Gemini API) and automatic metrics (ROUGE-L, BERTScore), and provided quantitative latency measurements [*tgXq, RsQN*].
- Clarified real-world applicability by emphasizing sample efficiency, OOD generalization, and modest inference overhead compared to training-based alternatives [*Lg6E*].

These additions and clarifications significantly strengthen the paper. We also thank the AC and SAC for their guidance and oversight throughout the review process.

---

### Decision · Action_Editor_9jyj · 2026-02-22

**Recommendation:** Reject

**Additional Comments:**

Strengths:

- The method is computationally efficient.

- It is sample-efficient, showing gains with very few annotated examples.

Reasons for Rejection:
- As noted by Reviewer RsQN and tgXq, the method is a close extension of existing semi-parametric approaches like kNN-LM, and its functional similarity to RAG limits its conceptual contribution.

- There is a persistent concern that the method relies on "nearest-neighbor answer carrying." The authors' clarification that RAD is a "calibration method" rather than a "reasoning method" helps define the scope, but reviewers felt this narrowed the paper's impact to a distributional correction trick rather than a robust solution for truthfulness.

**Audience:**

Yes

**Audience Explanation:**

The TMLR audience interested in RAG, and lightweight methods for hallucination reduction would find the RAD method noteworthy.

**Claims And Evidence:**

No

**Claims Explanation:**

While the authors provided extensive benchmarking across multiple datasets, significant concerns remain regarding the interpretation of the evidence. Reviewers noted that the performance gains might be attributed to "distributional exploitation" or "information leakage" from semantically similar examples in the small grounding space, rather than a genuine improvement in the model's truthful reasoning capabilities. The evidence does not sufficiently decouple the benefits of nearest-neighbor logit-shaping from simple template or answer matching.

**Resubmission Of Major Revision:**

The authors may consider submitting a major revision at a later time.